# Association between Pre-Existing Long-Term β-Blocker Therapy and the Outcomes of Sepsis-Associated Coagulopathy: A Retrospective Study

**DOI:** 10.3390/medicina58121843

**Published:** 2022-12-15

**Authors:** Ying Ma, Jie Ma, Jiong Yang

**Affiliations:** 1Department of Respiratory and Critical Care Medicine, Zhongnan Hospital of Wuhan University, Wuhan 430071, China; 2Department of Mathematics and Physics, North China Electric Power University–Baoding, Baoding 071003, China

**Keywords:** sepsis-associated coagulopathy, β-blocker, mortality, catecholamine, sepsis

## Abstract

*Background and Objectives:* Previous studies have suggested that long-term β-blocker therapy before sepsis is associated with reduced mortality. Sepsis-associated coagulopathy (SAC) remains a common complication in patients with sepsis and is associated with increased mortality. Adrenergic pathways are involved in the regulation of the coagulation system. Pre-existing long-term β-blocker therapy may have potentially beneficial effects on SAC and has yet to be well characterized. We aimed to assess the potential association between pre-existing long-term β-blocker therapy and the outcomes of patients with SAC. *Materials and Methods:* This study retrospectively screened the clinical data of adult patients with SAC admitted to the Intensive Care Unit (ICU) and respiratory ICU between May 2020 and October 2022. Patients with SAC who took any β-blocker for at least one year were considered pre-existing long-term β-blocker therapy. All enrolled patients were followed up for 28 days or until death. *Results:* Among the 228 SAC patients, 48 received long-term β-blocker therapy before septic episodes. Pre-existing long-term β-blocker therapy was associated with reduced vasopressor requirements and a decreased 28-day mortality (log-rank test: *p* = 0.041). In particular, long-term β-blocker therapy was related to substantially lower D-dimer levels and a trend of improved activated partial thromboplastin time in patients with SAC during initial ICU admission. Multivariable regression analysis showed that long-term β-blocker therapy was significantly and independently associated with a 28-day mortality among patients with SAC (adjusted odds ratio, 0.55; 95% confidence interval, (0.32–0.94); *p* = 0.030). *Conclusions:* Pre-existing long-term β-blocker therapy might be associated with reduced vasopressor requirements and a decreased 28-day mortality among patients with SAC, providing evidence for the protective effect of β-blockers against SAC in managing sepsis.

## 1. Introduction

Sepsis, an infection-induced serious condition generally defined as a life-threatening organ dysfunction, remains the most common cause of death in critically ill patients, even when prompt care is provided [1]. It results from a dysregulated host immune response to infection, often complicated by coagulopathy [2,3]. It is also well-known that coagulation abnormalities considerably contribute to developing multiple organ dysfunction and worse outcomes in sepsis [4]. Sepsis-associated coagulopathy (SAC) is characterized by a prolonged international normalized ratio (INR) and reduced platelet (PLT) count, which is associated with mortality [3,5]. Sepsis increases platelet reactivity, activates the coagulation system, and suppresses fibrinolysis [4]. Increased coagulation system activation results in extensive thrombus formation throughout the microvasculature and rapid consumption of coagulant factors, which may manifest as a prolonged plasma clotting time during sepsis [6]. These alterations in the coagulation pathway also cause decreased fibrinolysis, leading to disseminated intravascular coagulation, which results in microcirculation disturbance, promoting further inflammation and exacerbating organ dysfunction in sepsis [4,6]. Given the scourge of SAC in sepsis, several appropriate anticoagulant adjunctive treatments were used in clinical trials for managing this complication [7,8,9]. However, there are currently no effective interventions to provide beneficial effects on survival in patients with SAC [10].

Clinically, norepinephrine is the first-line vasopressor of choice for treating sepsis-related circulatory dysfunction because of its ability to induce vasoconstriction and its positive inotropic properties, despite the fact that excessive activation of the sympathetic nervous system may exert harmful effects on multi-organ functions, including cardiogenic, metabolic, immunological, and coagulopathic dysfunction [1,11,12]. Excess catecholamines may affect hemostasis by driving the coagulation–fibrinolysis balance toward hypercoagulability and by promoting platelet activation, aggregation and secretion, which translates into significantly accelerated blood clotting and an increased risk for thrombosis [12,13]. In light of this, the concept of ‘decatecholamisation’ has emerged, and β-blockers could represent a promising option due to their ability to reduce the need for administration of exogenous catecholamines and limit the overstimulation of endogenous catecholamine hormones for managing sepsis [14,15].

Notably, several recent studies have supported the concept discussed above. A previous clinical trial found that administering the β1-blocker esmolol in septic shock patients reduced norepinephrine requirements and improved the 28-day mortality [16]. Several retrospective studies have shown that premorbid β-blocker exposure to sepsis could exert its effect on reducing sepsis mortality [17,18]. Moreover, systematic reviews have also suggested the potential effect of a β-blocker therapy in septic patients on the 28-day mortality [19,20]. However, no retrospective cohort study has been conducted on the possible effects of pre-existing long-term β-blocker therapy on SAC. This study aims to evaluate the potential association between pre-existing long-term β-blocker therapy and SAC patients’ outcomes.

## 2. Materials and Methods

### 2.1. Study Design

This retrospective study was performed using the electronic medical record (EMR) systems of Zhongnan Hospital of Wuhan University. The study included patients with SAC admitted to the Intensive Care Unit (ICU) and respiratory ICU between May 2020 and October 2022. The study was approved by the ethics board of Zhongnan Hospital of Wuhan University (Approval Number: 2022152K), and informed consent was not required because of the retrospective review of the patients’ electronic medical records.

### 2.2. Study Patients

Critically ill septic patients aged ≥18 years diagnosed with SAC were included. The sepsis and septic shock diagnosis were in accordance with the Sepsis 3.0 criteria [1]. Patients with sepsis were eligible for inclusion if they had a known or suspected infection based on clinical data at admission and a Sequential Organ Failure Assessment (SOFA) score ≥2. Septic shock patients can be identified by the requirement of vasopressors in maintaining mean arterial pressure above 65 mmHg and serum lactate levels greater than 2 mmol/L despite adequate fluid resuscitation. Sepsis-associated coagulopathy was characterized by (1) an INR greater than 1.40 without other known etiology (e.g., anticoagulant therapy or chronic liver disease), and (2) a platelet count between 30 × 10^9^/L and 150 × 10^9^/L or greater than a 30% decrease in a platelet count within 24 h [6]. Patients were excluded from the study if they met the following criteria: premature death, lack of initial coagulation parameters, and a pre-existing history that affects the coagulation profile (malignancy, coagulopathy, or therapeutic anticoagulation).

### 2.3. Measurement of the Outcomes

The primary objective was to evaluate the association between pre-existing long-term beta-blocker therapy and SAC patients’ outcomes. The secondary outcome was to assess the differences in coagulation parameters between long-term β-blocker treated and untreated SAC patients.

### 2.4. Data Collection

Clinical data on ICU admission of all enrolled SAC patients were collected from the EMR system. In this study, 976 patients admitted for sepsis between May 2020 and October 2022, of which 273 met the inclusion criteria of SAC with 228 eventually being included. Patients with SAC who had taken any β-blocker for at least one year were considered to have a pre-existing long-term β-blocker therapy. Based on pre-existing long-term beta-blocker therapy data collected from the EMR system, patients were categorized into beta-blocker and no beta-blocker groups. Demographic information, comorbidities, source of infection, initial vital signs, blood laboratory tests, SOFA score, and main clinical outcomes, including the frequency of septic shock, mechanical ventilation, vasopressor requirements, and a 28-day mortality were collected. Vasopressor requirements were defined as the administration dose of norepinephrine to maintain mean arterial pressure above 65 mmHg in the first 24 h of ICU admission. Vasopressin and epinephrine doses were converted to their norepinephrine equivalents [21].

### 2.5. Statistical Analysis

No sample size calculation was performed because this was a retrospective, exploratory study. Continuous data are reported as mean ± standard deviation (SD) or median (interquartile range (IQR)), as appropriate. Categorical data are expressed as numbers and percentages. Continuous data were compared using the Mann–Whitney test, and categorical variables were compared using the chi-square test. The Kaplan–Meier analysis was performed for 28-day mortality, and the differences were evaluated using the log-rank test. The independent risk factors associated with 28-day mortality among SAC patients were evaluated using the cox proportional-hazards regression analysis. Variables with significant associations in the univariable analysis were included in a final multivariable regression model. Statistical significance was set at *p* < 0.05. Data were analyzed using SPSS version 23 (IBM, Chicago, IL, USA) and plotted using GraphPad PRISM version 5.3 (San Diego, CA, USA).

## 3. Results

### 3.1. Patient Characteristics

Two hundred seventy-three SAC patients were evaluated during the study period, and 45 were excluded due to premature death (*n* = 5), lack of initial coagulation parameters (*n* = 7), and a pre-existing history that affects the coagulation profile (malignancy, coagulopathy, or therapeutic anticoagulation) (*n* = 33) (Appendix A). A retrospective cohort study was performed on 228 eligible patients. Of these, 48 received long-term β-blocker treatment, and 180 did not. Among the patients who received β-blocker treatment, 41 took the selective β1-blocker metoprolol, and 7 took bisoprolol. Patient demographics and baseline clinical characteristics are summarized in Table 1. Hypertension, diabetes, and cerebrovascular diseases were more prevalent in patients with long-term β-blocker treatment. SAC patients treated with long-term β-blocker and those untreated had similar sources of infection. Both patient groups had significantly elevated white blood cell counts and percentage of neutrophils, and a marked decrease in the percentage of lymphocytes during initial admission, whereas no significant differences were observed among the beta-blocker and no beta-blocker groups. During initial ICU admission, patients with SAC had a significantly reduced PLT and a prolonged INR when compared with the normal range. Similarly, no significant differences in albumin (ALB), glucose, bilirubin, and creatinine were observed between long-term β-blocker treated and untreated patients. Patients with SAC presented significantly elevated SOFA scores during initial ICU admission, and untreated patients had higher SOFA scores than those treated with β-blockers (11.5 ± 4.6 vs. 9.9 ± 4.4, *p* = 0.034).

### 3.2. Pre-Existing Long-Term β-Blocker Therapy and Clinical Outcomes

Compared with untreated patients, long-term β-blocker therapy before septic episodes was associated with a lower incidence of septic shock and a lower usage of mechanical ventilation in patients with SAC (septic shock: 64.6% vs. 78.9%, *p* = 0.040; mechanical ventilation: 39.6% vs. 56.7%, *p* = 0.035), as shown in Table 2. SAC patients treated with long-term β-blocker treatment had reduced vasopressor requirements when compared to untreated patients (0.11 (0–0.32) μg/kg/min vs. 0.32 (0.06–1.48) μg/kg/min, *p* < 0.001). All enrolled patients were followed up for 28-day mortality. The survival curve of patients with SAC showed that pre-existing long-term β-blocker therapy was associated with a decreased 28-day mortality (log-rank test: *p* = 0.041) (Figure 1).

### 3.3. Changes in Markers of Coagulation Function

Markers of coagulation function on day 4 of ICU admission were not performed on all enrolled patients because of premature death and variations in care between clinicians. During the study period, the number of patients with SAC with coagulation function parameters on days 1 and 4 of ICU admission was 228 and 153, respectively (Table 3). Patients with SAC presented with a significantly more prolonged prothrombin time (PT) and activated partial thromboplastin time (APTT) among the beta-blocker and no beta-blocker groups on day 1 of ICU admission. There was no significant difference in PT in patients treated with long-term β-blocker therapy when compared to those untreated. Whereas patients with SAC demonstrated a relatively shorter APTT in the beta-blocker group when compared with those in the no beta-blocker group on day 1 of ICU admission, but the difference did not reach statistical significance (35.8 (32.8–39.0) s vs. 38.1 (32.6–48.5) s, *p* = 0.057). In contrast, initial thrombin time (TT) and fibrinogen (FIB) did not differ significantly between long-term β-blocker treated and untreated patients. Initial plasma levels of D-dimer (DD) were significantly elevated in both groups of patients with SAC. When compared with patients treated with long-term β-blocker therapy, patients in the no beta-blocker group had significantly higher levels of DD on day 1 of ICU admission (3138 (1428–5923) ng/mL vs. 2474 (993–3571) ng/mL, *p* = 0.016). Patients with SAC seemed to show relative improvements in the markers of coagulation function on day 4 of ICU admission. No statistically significant differences were noted in the PT, APTT, TT, and FIB between the two groups. Pre-existing long-term β-blocker therapy was related to a reduced trend of DD levels in patients with SAC, but the difference was not significant (2009 (1225–3158) ng/mL vs. 2722 (1383–4286) ng/mL, *p* = 0.131).

### 3.4. Pre-Existing Long-Term β-Blocker Therapy and 28-Day Mortality

Table 4 shows the univariate and multivariate cox proportional analysis results that evaluated the variables associated with 28-day mortality among patients with SAC. The univariate cox regression analysis showed that β-blocker therapy was associated with 28-day mortality (hazard ratio, 0.59; 95% confidence interval (CI), (0.35–1.00); *p* = 0.048). In the multivariate regression analysis model, β-blocker therapy was significantly and independently associated with 28-day mortality among patients with SAC (adjusted hazard ratio, 0.55; 95% CI, (0.32–0.94); *p* = 0.030). In addition, coronary heart disease, fibrinogen, and the SOFA score were also independently associated with SAC-related mortality.

## 4. Discussion

This retrospective study showed that pre-existing long-term β-blocker therapy might be associated with a decreased 28-day mortality in patients with SAC. It is the first time that the impact of pre-existing long-term β-blocker therapy on coagulation disorders in sepsis has been evaluated. Long-term β-blocker therapy before septic episodes was related to reduced vasopressor requirements, substantially lower DD levels, and a trend of improved APTT in patients with SAC during initial ICU admission. Overall, our findings highlight the importance of understanding the potential association between pre-existing long-term β-blocker therapy and SAC and provides preliminary evidence that encourages more prospective studies to consider the potential appropriate adjunctive therapy of β-blocker use against SAC in managing sepsis.

β-blockers might be a promising therapeutic strategy in the management of sepsis due to their ability to reduce the requirements of exogenous catecholamines. In this study, we found that long-term β-blocker therapy was significantly and independently associated with 28-day mortality among patients with SAC. Interestingly, among the SAC patients in our cohort, despite old age and more complications in long-term β-blocker treated patients, they had lower SOFA scores than untreated patients. One possible explanation is the vasopressor requirements during sepsis. Over-stimulation with catecholamines may exacerbate immune, metabolic and coagulation disorders, and organ failure during sepsis [22,23]. A higher vasopressor load in septic shock patients is associated with an increased in-hospital mortality [22]. As discussed above, Morelli et al. found that the administration of esmolol in septic shock patients resulted in a reduction in norepinephrine requirements [14]. A previous experimental study showed that the beneficial effects of esmolol may be attributed to the improvement of vascular function [24]. In agreement with previous retrospective studies [25,26], we found that long-term β-blocker therapy before septic episodes might be associated with lower vasopressor requirements in SAC patients. The lower vasopressor requirement also limits the potential adverse effects of high-dose catecholamines on sepsis patients [25]. Hence, we postulated that the protective effect of pre-existing long-term β-blocker therapy might be related to its role in increasing patients’ tolerance to the excessive catecholamine surge during sepsis and contributing to the improvement of coagulation dysfunction.

Moreover, inflammation was reported to play an essential role in atherosclerotic plaque formation, vulnerability, and rupture, increasing cardiovascular risks [27]. The coagulation disorder associated with acute infection further increases the risk of plaque rupture and thrombosis. Similar to prior research [28], our analyses showed that coronary heart disease was independently associated with 28-day mortality among SAC patients. SAC patients with coronary heart disease should be closely monitored to further reduce adverse cardiovascular events. Fibrinogen is the key component involved in SAC. Hypofibrinogenemia was independently associated with the increased risk of in-hospital mortality in adult patients with severe sepsis [29]. In our study, fibrinogen was an independent risk factor for SAC-related mortality, which might allow clinicians to better risk-stratify SAC patients in managing sepsis. Together, these findings might have potential implications for the management of SAC. However, future prospective studies are needed to examine whether these mortality risk factors have a prognostic value for SAC patients.

Adrenergic pathways are associated with the coagulation system. The β-adrenergic receptors are expressed on the surface of platelets and can modulate platelet function [30]. Increased circulating epinephrine during sepsis increases platelet reactivity via adrenergic receptors [31]. The β2-adrenergic pathway may suppress platelet aggregation via cAMP stimulation [30]. Although the underlying mechanisms remain unclear, β1-blocker atenolol appeared to induce a decreased in vitro platelet adhesion in patients with hypertension [32]. Furthermore, β1-adrenergic stimulation suppresses fibrinolysis by reducing prostacyclin synthesis [30], whereas β2-adrenergic stimulation promotes tissue plasminogen activator release, leading to an enhanced fibrinolytic activity [33]. In light of this information, it can be predicted that pre-existing long-term β-blocker therapy might have potential protective effects against SAC in patients with sepsis or septic shock. The present study showed that long-term β-blocker therapy was related to substantially lower levels of DD and a trend of improved APTT in patients with SAC, indicating the potential benefits of β-blockers on coagulation function during sepsis. 

However, the effects of β1 and β2 adrenergic pathways might be contradictory. β2-blockers might counteract the β2-induced decrease in platelet activation and improvement in fibrinolytic activity [23], which could be detrimental during sepsis. Therefore, the β1-blockers could have a beneficial effect on fibrinolytic status through relative β2-adrenergic activation. In addition, esmolol potentially improved microvascular circulation and fibrinolytic activity during septic shock [34]. This study was not designed further to explore the impacts of different β-blocker classes on SAC; hence, we could not compare the specific effects of β1-blockers and β2-blockers because of the small cohort population. In a retrospective study, Kuo et al. found that a reduced ICU mortality was associated with premorbid β1-selective exposure [25]. Furthermore, a recent multicenter cohort study showed that premorbid non-cardioselective β-blocker exposure might be related to a decreased adjusted ICU and hospital mortality in sepsis [26]. Therefore, the causal relationship between premorbid β1-selective blocker exposure and mortality during sepsis could not be determined. Further prospective clinical trials are needed to explore the therapeutic potential of non-cardioselective β-blocker and β1-selective blockers against SAC in managing sepsis.

Several limitations of this study should be considered when interpreting our findings. First, it was a retrospective, single-center study with a relatively small sample size. Potential selection bias (e.g., previous history, disease severity, and old age) challenge the interpretation from establishing definitive causal associations. Since long-term β-blocker treated patients were older, had more chronic diseases, and were taking more medications, some treated patients with lower SOFA scores were more likely to be admitted to the ICU and receive improved care than untreated patients. Here, we provided support for a potential relationship between pre-existing long-term β-blocker therapy and the outcomes of SAC patients. Second, we only collected data beginning from the ICU admission, considering that many sepsis patients might be treated partially in the emergency department or the community EDs before transfer to the ICU. Pre-existing complications with enrolled patients contribute to more severe sepsis and septic shock than sepsis patients in general community hospitals, which may have caused bias. Therefore, the results may not be generalizable to other locations. Hence, future extensive prospective studies are required to validate these findings. Third, the retrospective analysis was based solely on the history of β-blocker exposure in medical records. Other unaccounted confounders, such as difficulty following medication instructions, pre-existing clinical scenarios of anticoagulants, continuing long-term β-blocker therapy during sepsis, or sepsis-associated impairment of liver synthesis, may also play a role in coagulation disorders during sepsis. Fourth, because of the small sample size, we could not further assess the potential impacts of different β-blocker classes on the outcomes in SAC patients. Therefore, large-scale prospective multicenter studies are warranted to assess the impact of contributing factors to the coagulation cascade in the entire course of sepsis and further evaluate the therapeutic potential of β-blockers in sepsis.

## 5. Conclusions

In conclusion, our findings suggest that pre-existing long-term β-blocker therapy might be associated with a decreased 28-day mortality among patients with sepsis-associated coagulopathy. Here, we set the stage for further prospective clinical trials to consider β-blockers as a potentially appropriate adjunctive therapy against sepsis-associated coagulopathy in managing sepsis.

## Figures and Tables

**Figure 1 medicina-58-01843-f001:**
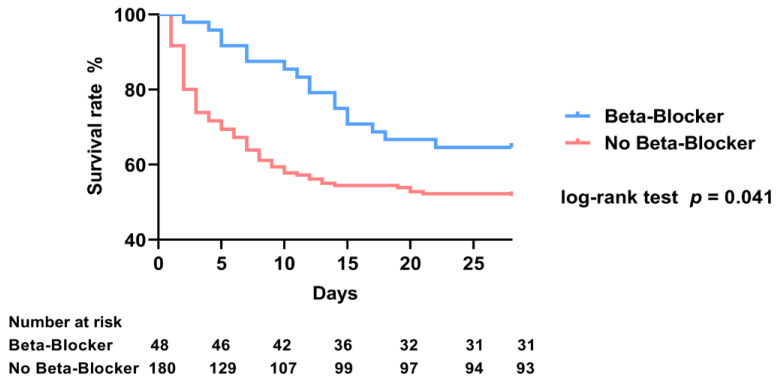
Kaplan–Meier survival curves of patients with sepsis-associated coagulopathy grouped by the status of pre-existing long-term β-blocker therapy. Beta-Blocker: patients with sepsis-associated coagulopathy treated with long-term β-blocker therapy before septic episodes. No Beta-Blocker: patients with sepsis-associated coagulopathy treated without long-term β-blocker therapy before septic episodes.

**Table 1 medicina-58-01843-t001:** Demographic and baseline clinical characteristics of the study population.

Variables	Total *n* = 228	Beta-Blocker *n* = 48	No Beta-Blocker *n* = 180	*p* Value
Age, years, mean ± SD	66.1 ± 15.7	72.8 ± 12.3	64.3 ± 16.1	0.001
Male sex, *n* (%)	139 (61.0%)	25 (52.1%)	114 (63.3%)	0.156
Comorbidities				
Hypertension, *n* (%)	92 (40.4%)	39 (81.3%)	53 (29.4%)	<0.001
Diabetes, *n* (%)	48 (21.1%)	16 (33.3%)	32 (17.8%)	0.019
Chronic lung disease, *n* (%)	26 (11.4%)	4 (8.3%)	22 (12.2%)	0.451
Coronary heart disease, *n* (%)	49 (21.5%)	20 (41.7%)	29 (16.1%)	<0.001
Cerebrovascular disease, *n* (%)	44 (19.3%)	13 (27.1%)	31 (17.2%)	0.124
Others, *n* (%)	64 (28.1%)	13 (27.1%)	51 (28.3%)	0.864
Source of infection				
Pulmonary infection, *n* (%)	106 (46.5%)	17 (35.4%)	89 (49.4%)	0.083
Intraabdominal infection, *n* (%)	74 (32.5%)	19 (39.6%)	55 (30.6%)	0.235
Urinary infections, *n* (%)	44 (19.3%)	13 (27.1%)	31 (17.2%)	0.124
Soft tissue infection, *n* (%)	14 (6.1%)	3 (6.3%)	11 (6.1%)	0.972
Bacteraemia, *n* (%)	20 (8.3%)	4 (8.3%)	16 (8.8%)	0.904
Other sources, *n* (%)	6 (2.6%)	2 (4.2%)	4 (2.2%)	0.455
Vital lab data				
WBC (10^9^/L), median (IQR)	11.5 (6.9–17.1)	11.6 (8.7–16.3)	11.4 (6.5–17.7)	0.826
Neutrophils (%), median (IQR)	91.0 (83.6–94.0)	91.2 (84.4–94.0)	91.0 (82.6–94.0)	0.961
Lymphocytes (%), median (IQR)	4.3 (2.6–8.8)	4.4 (2.2–8.0)	4.2 (2.7–9.8)	0.462
Monocytes (%), median (IQR)	3.7 (2.4–6.1)	4.2 (3.0–6.3)	3.6 (2.2–6.0)	0.209
Platelet (10^9^/L), median (IQR)	73.0 (42.0–103.0)	74.5 (54.0–107.8)	73.0 (41.0–99.3)	0.354
INR, median (IQR)	1.61 (1.47–1.99)	1.60 (1.46–1.94)	1.62 (1.48–2.00)	0.543
HGB (g/L), median (IQR)	98.0 (81.3–114.0)	96.0 (86.5–113.0)	98.5 (79.0–114.3)	0.96
ALB (g/L), median (IQR)	27.5 (23.9–30.6)	28.3 (26.5–30.5)	27.0 (23.6–30.6)	0.156
Glucose (mmol/L), median (IQR)	6.6 (5.3–8.9)	6.7 (5.8–9.0)	6.5 (5.1–8.9)	0.186
Bilirubin (μmol/L), median (IQR)	33.1 (17.3–65.2)	28.2 (13.2–64.0)	33.9 (18.1–65.2)	0.391
Creatinine (μmol/L), median (IQR)	167.1 (96.1–256.4)	177.8 (113.4–252.0)	163.4 (85.3–256.4)	0.314
SOFA score, mean ± SD	11.2 ± 4.6	9.9 ± 4.4	11.5 ± 4.6	0.034

WBC: White Blood Cells; PLT: Platelet; INR: international normalized ratio; HGB: hemoglobin; ALB: albumin; SOFA score: Sequential Organ Failure Assessment score; SD: standard deviation; IQR: interquartile range. Beta-Blocker: patients with sepsis-associated coagulopathy treated with long-term β-blocker therapy before septic episodes. No Beta-Blocker: patients with sepsis-associated coagulopathy treated without long-term β-blocker therapy before septic episodes.

**Table 2 medicina-58-01843-t002:** Clinical outcomes of the study population.

Variables	Total *n* = 228	Beta-Blocker *n* = 48	No Beta-Blocker *n* = 180	*p* Value
Septic shock, *n* (%)	173 (75.9%)	31 (64.6%)	142 (78.9%)	0.040
Mechanical ventilation, *n* (%)	121 (53.1%)	19 (39.6%)	102 (56.7%)	0.035
Norepinephrine equivalents total * (μg/kg/min), median (IQR)	0.24 (0–1.20)	0.11 (0–0.32)	0.32 (0.06–1.48)	<0.001
Mortality on day 28, *n* (%)	104 (45.6%)	17 (35.4%)	87 (48.3%)	0.110

* Vasopressin and epinephrine doses were converted to their norepinephrine equivalents. IQR: interquartile range. Beta-Blocker: patients with sepsis-associated coagulopathy treated with long-term β-blocker therapy before septic episodes. No Beta-Blocker: patients with sepsis-associated coagulopathy treated without long-term β-blocker therapy before septic episodes.

**Table 3 medicina-58-01843-t003:** Changes in markers of coagulation function.

Variables	Day 1 of ICU Admission		*p* Value	Day 4 of ICU Admission		*p* Value
	Beta-Blocker (*n* = 48)	No Beta-Blocker (*n* = 180)		Beta-Blocker (*n* = 37)	No Beta-Blocker (*n* = 116)	
PT (s),	17.5	17.8	0.511	14	14.6	0.818
median (IQR)	(15.8–21.4)	(16.1–21.8)		(13.0–17.3)	(13.0–17.4)	
APTT (s),	35.8	38.1	0.057	33.9	33.4	0.777
median (IQR)	(32.8–39.0)	(32.6–48.5)		(30.1–38.2)	(29.8–41.2)	
TT (s),	15.2	15.4	0.554	15.9	16	0.516
median (IQR)	(13.8–17.2)	(14.0–17.8)		(14.0–16.6)	(14.0–18.2)	
FIB (mg/dL),	394	370.5	0.604	334	337	0.895
median (IQR)	(283.5–431.8)	(243.3–452.3)		(219.0–380.0)	(228.5–432.0)	
DD (ng/mL),	2474	3138	0.016	2009	2722	0.131
median (IQR)	(993–3571)	(1428–5923)		(1225–3158)	(1383–4286)	

PT: prothrombin time; APTT: activated partial thromboplastin time; TT: thrombin time; FIB: fibrinogen; DD: D-dimer; IQR: interquartile range. Beta-Blocker: patients with sepsis-associated coagulopathy treated with long-term β-blocker therapy before septic episodes. No Beta-Blocker: patients with sepsis-associated coagulopathy treated without long-term β-blocker therapy before septic episodes.

**Table 4 medicina-58-01843-t004:** Cox proportional hazard analysis of risk for 28-day mortality in patients with sepsis-associated coagulopathy in univariate and multivariate models.

Variables	Univariate		Multivariate *	
	Hazard Ratio	*p* Value	Adjusted Hazard Ratio	*p* Value
Beta-Blocker	0.59 (0.35–1.00)	0.048	0.55 (0.32–0.94)	0.03
Age	1.01 (1.00–1.02)	0.259	-	-
Comorbidities				
Hypertension	1.41 (0.96–2.08)	0.078	-	-
Diabetes	0.98 (0.61–1.57)	0.943	-	-
Chronic lung disease	1.70 (0.99–2.90)	0.051	-	-
Coronary heart disease	1.61 (1.05–2.47)	0.03	1.83 (1.18–2.83)	0.007
Cerebrovascular disease	1.10 (0.69–1.76)	0.687	-	-
Source of infection				
Pulmonary infection	1.21 (0.83–1.79)	0.319	-	-
Intraabdominal infection	0.97 (0.64–1.47)	0.884	-	-
Urinary infections	0.75 (0.44–1.25)	0.27	-	-
WBC	0.99 (0.96–1.01)	0.195	-	-
Lymphocytes	1.02 (1.01–1.03)	<0.001	1.01 (1.00–1.03)	0.153
PLT	0.99 (0.99–1.00)	0.008	1.00 (1.00–1.01)	0.663
INR	1.41 (1.14–1.74)	0.001	1.27 (0.97–1.67)	0.087
FIB	1.00 (1.00–1.00)	<0.001	1.00 (1.00–1.00)	0.04
DD	1.00 (1.00–1.00)	0.474	-	-
ALB	0.93 (0.90–0.97)	0.001	0.97 (0.93–1.01)	0.178
Glucose	1.02 (0.99–1.06)	0.246	-	-
Bilirubin	1.00 (1.00–1.00)	0.435	-	-
Creatinine	1.00 (1.00–1.00)	0.081	-	-
SOFA score	1.25 (1.19–1.31)	<0.001	1.22 (1.15–1.28)	<0.001

* Adjusted for variables with *p* < 0.05 in the univariate analysis. WBC: White Blood Cells; PLT: Platelet; INR: international normalized ratio; FIB: fibrinogen; DD: D-dimer; ALB: albumin.

## Data Availability

The data presented in this study are available on request from the corresponding author.

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
