# Peer review of "Association between Pre-Existing Long-Term β-Blocker Therapy and the Outcomes of Sepsis-Associated Coagulopathy: A Retrospective Study"

_medicina, 2022, doi:10.3390/medicina58121843_

Round 1

Reviewer 1 Report

I appreciate the opportunity to review the manuscript titled “Association between Pre-existing Long-term β-Blocker Therapy and the Outcomes of Sepsis-Associated Coagulopathy: A Retrospective Study” by Ma. et. al. The authors assessed impact of long term beta-blocker therapy on outcomes of sepsis-associated coagulopathy. My suggestion included in the word file are with the hope to strengthen the manuscript.

Author Response

Dear reviewer,

We appreciate you and the reviewers for your precious time reviewing our paper and providing valuable comments. Your valuable and insightful comments led to possible improvements in the current version. The authors have read the comments carefully and tried our best to address them one by one. We hope the manuscript meets your high standards after careful revisions. The authors welcome further constructive comments, if any. 

Below we provide the response. Please see the attachment.

Sincerely,

Jiong Yang

[email protected]

Reviewer 2 Report

The authors present an innovative retrospective study regarding a possible effect of pre-existing long term usage of beta-blockers. They found a lower 28 day-mortality and lower vasopressor requirements in patients with beta-blocker-usage. These results are interesting, however several major issues need to be clarified. 1) The most important limitation is the fact that patients without beta-blockers were older and sicker. This is not discussed. 2) the information regarding anticoagulation would be nice. 3) seemingly an error between the Kaplan Meier curves and the tables occurred (different numbers of patients stated). 4) all results should be discussed.

Please find more detaield information below:

I´m missing information on the anticoagulation of the patients as this could interfere with the findings of this retrospective study.

Abstract:

Line 25: “relatively improved activated partial thromboplastin time in patients with SAC during initial ICU adimission”. This was only a trend, but not significant.

Introduction:

Line 42: PLT count or platelets

Line 43: is SAC really a deadly complication? Or only the potential  consequences like pulmonary or cerebral bleeding?

Results:

Line 114: the number for each reason of the excluded patients would have be nice, like premature deaths (n = …), ...

Line 118: these are both cardioselective, aren´t they?

Fig 1 states 94 patients surviving 28 days in the no beta-blocker-group. Given the numbers in table 2, there should survive only 93 patients after 28 days (180 – 47 = 93). Moreover, in table 3 37 patients survived until day 5, which is in contrast to fig 1 with 46 patients surviving at day 4 (same with non beta-blocker-group: 116 in table 3 vs. 129 in fig 1).

Discussion:

Line 208f: patients with beta-blockers showed only a trend towards a lower APTT. However, this was not significant. Please change this part of the sentence.

Line 217ff: impairment of liver synthesis may also promote to a prolonged plasma clotting time during sepsis.

Line 229ff: the association between long term beta-blocker usage and lower vasopressor requirement may be due to the fact that patients with no beta-blockers were the sicker ones as SOFA scores were higher. This major issue should be discussed.

The fascinating question is why the patients without beta-blocker were sicker as these patients showed a lower portion of hypertension, diabetes and coronary heart disease.

Line 233ff: this conclusion is overly speculative given the said above

Line 247ff: again, this effect could be due to the less diseased beta-blocker-group

Furthermore, I´m missing an interpretation of the rest of the results, ie. patients with beta-blocker being older and sicker. Moreover, multivariate analysis showed an association of lower albumin, coronary heart disease and fibrinogen. This should be discussed as well to provide the readership a thorough understanding of the results.

Minor comments:

line 25: check spelling (adimission)

Author Response

Dear reviewer,

We appreciate you for your precious time reviewing our paper and providing valuable comments. Your valuable and insightful comments led to possible improvements in the current version. The authors have carefully considered the comments and tried our best to address every one of them. We hope the manuscript meets your high standards after careful revisions. The authors welcome further constructive comments, if any.

Below we provide the point-by-point responses. Please see the attachment

Round 2

Reviewer 1 Report

Thank you so much for taking all my suggestions into consideration and answering all my questions appropriately. 

Reviewer 2 Report

Thank you for incorporating my comments to the manuscript. Yes, I agree on the impact of SAC on in-hospital mortality as demonstrated by Lyons and co-workers. However, as SAC increases mortality, this doesn´t have to be a death sentence for a patient. "Deadly" just senses very dramatic to my mind (line 47). Anyhow, you may leave this sentence.

The major revision made and your thorough answers improved the manuscript singificantly. Therefore I feel the mansucript suitable for publication now.